# Pain Course after Total Knee Arthroplasty within a Standardized Pain Management Concept: A Prospective Observational Study

**DOI:** 10.3390/jcm11237204

**Published:** 2022-12-04

**Authors:** Melanie Schindler, Stephanie Schmitz, Jan Reinhard, Petra Jansen, Joachim Grifka, Achim Benditz

**Affiliations:** 1Department of Orthopedics, University Medical Center Regensburg, Asklepios Klinikum Bad Abbach, 93077 Bad Abbach, Bavaria, Germany; 2Department of Sport Science, University of Regensburg, 93053 Regensburg, Bavaria, Germany; 3Department of Orthopedics, Klinikum Fichtelgebirge, 95615 Marktredwitz, Bavaria, Germany

**Keywords:** total knee arthroplasty, postoperative pain, numeric rating scale, influencing factors

## Abstract

Background: Joint replacement surgeries have been known to be some of the most painful surgical procedures. Therefore, the options for postoperative pain management are of great importance for patients undergoing total knee arthroplasty (TKA). Despite successful surgery, up to 30% of the patients are not satisfied after the operation. The aim of this study is to assess pain development within the first 4 weeks after TKA in order to gain a better understanding and detect possible influencing factors. Methods: A total of 103 patients were included in this prospective cohort study. Postoperative pain was indicated using a numeric rating scale (NRS). Furthermore, demographic data and perioperative parameters were correlated with the reported postoperative pain. Results: The evaluation of postoperative pain scores showed a constant decrease in the first postoperative week (mean NRS score of 5.8 on day 1 to a mean NRS score of 4.6 on day 8). On day 9, the pain increased again. Thereafter, a continuous decrease in pain intensity from day 10 on was noted (continuous to a mean NRS score of 3.0 on day 29). A significant association was found between postoperative pain intensity and gender, body mass index (BMI), and preoperative leg axis. Conclusions: The increasing pain score after the first postoperative week is most likely due to more intensive mobilization and physiotherapy in the rehabilitation department. Patients that were female, had a low BMI, and a preoperative valgus leg axis showed a significantly higher postoperative pain scores. Pain management should consider these results in the future to improve patient satisfaction in the postoperative course after TKA.

## 1. Introduction

Knee osteoarthritis causes pain and limited mobility. Hence, it is the main indication for total knee arthroplasty (TKA). TKA relieves pain, improves mobility, and thus increases quality of life. In Germany, primary TKA is one of the most frequently performed surgical procedures [1]. The total number of TKA procedures in Germany is expected to increase by 45%, from 168,772 procedures in 2016 to 244,714 procedures in 2040 [2]. As a result, it is all the more important that this stressful surgical procedure is successful for the patients. Surgical procedures are influenced by many factors, including patient and surgeon preferences. Pain is the most important factor in patient satisfaction [3]. Most patients achieve postoperative pain reduction with a good clinical outcome [1]. 10–20% of patients are dissatisfied with the surgical outcome and report persistent chronic pain postoperatively (CPSP) [4]. This can lead to delayed mobilization, a longer duration of hospitalization, and thus higher costs for the health care system. Therefore, multidisciplinary pain management is of high significance. It is crucial to have a better understanding of this dissatisfaction and the factors that influence it. Patients with early postoperative persistent pain had a lower chance of being pain-free after one year than patients who reported no or only little pain. [5]. A detailed assessment of the postoperative pain course with a pain curve has not been performed up to now. The early postoperative phase and the rehabilitation phase both represent a particular challenge for patients and their reintegration.

Prolonged postoperative pain leads to increased consumption of analgesics and a longer rehabilitation stay. Therefore, the aim of this study is to evaluate postoperative pain development and detect possible factors influencing postoperative pain after TKA.

## 2. Material and Methods

This work is a prospective study of a single center of orthopedic surgery at a university hospital, including patients undergoing primary TKA between October 2020 and July 2021. The patients were enrolled on the day of preoperative preparation, which in our department usually takes place a few days before the surgical performance.

Patients received cemented PFC Sigma (Depuy Synthes, Warsaw, IN, USA) or cemented nickel-free NexGen^®^ knee prostheses (Zimmer Biomet Inc., Warsaw, IN, USA). Patellar resection arthroplasty with circumpatellar electrocautery and osteophyte removal was performed on all of the patients. Patellar resurfacing was not performed. Patients who received primary TKA, anesthesia via peripheral nerve block, sedation with propofol, and inpatient rehabilitation in our department were included in the study. The follow-up for patients became easier as the rehabilitation treatment was standardized. Patients with chronic pain syndromes preoperatively and/or an intraoperative change to general anesthesia were excluded. 

A standardized pain management regimen was given to all patients: Preoperatively, patients were given 7.5 mg of midazolam orally one hour before surgery. The psoas compartment block was performed with 20 mL of ropivacaine 0.75% and the ischiatic nerve proximal dorsal block (transgluteal) with 20 mL of prilocaine 1%. The peripheral nerve block was placed using neurostimulation, and the feedback was expected to be a twitch of the leg. During surgery, patients were sedated with propofol. In the intermediate care unit, 10 mL of ropivacaine 0.75% were administered to the patients via the psoas block at regular intervals during the first 12 h after surgery. Furthermore, patients use the pain catheter at 45 min intervals with 10 mL of ropivacaine 0.75% if needed. In cases of severe pain, the ischiatic nerve block was maintained with ropivacaine 0.2 6 mL/hr.

The standard oral analgesic medication, which was also given during the analgesic therapy via catheter, was metamizole 500 mg four times daily and ibuprofen 600 mg three times daily. In case of pain exacerbation, tramadol 100 mg (40 gtt) was provided, which could be repeated after 30 min when the NRS was 3–6. Also, oxycodone 20 mg could be given and repeated after 1 h in the case of an NRS of 7–10. If the patient used all therapy options, the standard analgesic medication was adjusted. Cold packs were also provided for the knee. Full weight bearing with crutches was allowed directly after surgery. 

The preoperative clinical status and the results one week and four weeks postoperatively were evaluated according to the Knee Society Score and Function Score (KSS and FS) [6].

A whole-leg radiograph was performed preoperatively and a few days after surgery. The measured radiographic parameters included the anatomical axis of the leg. It connects the anatomical femoral axis with the anatomical tibial axis and forms a physiological angle of 5° to 10° valgus. A positive degree value corresponds to a valgus position, a negative one to a varus position.

All patients documented their postoperative pain four times a day (morning, lunchtime, evening, and nighttime) and the maximum pain of the day using the numerous rating scale (NRS 0 = no pain; 10 = worst imaginable pain). In our department, the patients received physiotherapy once a day, including continuous passive motion (CPM) therapy. They got an intense rehabilitation program during the following stationary rehabilitation. 

The study was approved by the local ethics committee (16-101-0204). Information was supplied to all potential patients, and participation was voluntary. A written informed consent was received from every subject.

IBM SPSS Statistics 25 (IBM Corp., Armonk, NY, USA) was used for analysis. Demographics and clinical characteristics were presented as means and standard deviations. Predictors of postoperative pain were analyzed using linear regression models. Leg axis and function scores were evaluated using paired t-tests. A *p*-value < 0.05 was considered as statistically significant. 

## 3. Results

Initially, 139 patients were included in the study. 15% of the initial data could not be used because of incomplete/missing pain sheets (n = 7), discontinuation of rehabilitation treatment due to SARS-CoV-2 infection/contact (n = 8), or a second surgical procedure on account of a complication (n = 3). The complications that led to a revision surgery included wound healing disorder (n = 1), early infection (n = 1), and arthrofibrosis (n = 1). Finally, a participation rate of 85% (n = 103) was achieved.

The mean age of patients was 66.5 ± 8.7 years. Most of the patients were female (n = 56, 54.4%). According to the classification of the World Health Organization (WHO), 8.7% of the patients were of normal weight, 34% were pre-obese, and 57.3% were obese. Indications for performing TKA were osteoarthritis (88.3%) and post-traumatic osteoarthritis (11.7%). In general, 29.1% underwent a knee arthroscopy preoperatively and 32% had already a total hip or contralateral knee arthroplasty before. One third of the patients (35%) took painkillers daily, 35.9% casually, and 29.1% of the patients did not take any painkillers preoperatively. Most patients (68%) had an ASA score (American Society of Anesthesiologists) of 2, 22.4% had an ASA score of 3, and 9.7% had an ASA score of 1.

The mean duration of surgery was 82.7 ± 18.6 min (minimum 47, maximum 150). 84.5% received a cemented PFC Sigma total knee arthroplasty, and 15.5% received a cemented nickel-free NexGen implant. 61.2% of the operations were computer-assisted, and 38.8% were conventional TKAs. 

The pain catheter was removed at day 2.7 + 0.71 (min 1, max 5 days) on average. 

The mean anatomical axis of the leg showed a significant difference from 4.3 ± 7.2° (min −1°, max 25°) preoperatively to 7.2 ± 3.4° (min −2°, max 16°) postoperatively (*p* = 0.001).

The clinical outcome showed a preoperative KSS of 46 ± 15 points and FS of 56 ± 16 points, 1 week postoperatively 61 ± 16 (*p* = 0.001) and 41 ± 17 (*p* = 0.001), and after 4 weeks 69 ± 17 (*p* = 0.001) and 55 ± 11 (*p* = 0.735). Accordingly, the improvement was significant even without the last FS.

In the following postoperative period, the mean pain score was measured on days 1 to 29 (Table 1; Figure 1). The maximum and minimum pain of the day, documented by the patients, were then evaluated (Figure 2). In general, influencing factors were sex, BMI, and anatomical axis. Female gender (Figure 3), low BMI, and valgus leg axis showed a significant correlation with more severe postoperative pain scores. In contrast, age, ASA score, surgical duration, KS score, and FS score did not influence the pain score (Table 1).

## 4. Discussion

### 4.1. Postoperative Pain Course

Overall, postoperative pain decreases significantly after TKA. In the first postoperative week, the lowest pain scores were on day 8 with an NRS of 4.6. This increased to 4.8 on day 9. The reason for the increase in pain progression may be related to the start of the intensive rehabilitation program. Subsequently, there was a constant decrease in the pain level from day 10 to day 29. Another study showed a comparable mean maximum pain of 5.44 ± 1.83 on the first postoperative day [7]. Here, the NSR was 5.8 ± 2.8. An increase from the middle of the first postoperative week to the end of the week had already been observed in another study [8].

An identical study design has already been published for primary total hip arthroplasty. They described the postoperative pain over a course of four weeks as well as possible factors influencing pain intensity after primary total hip arthroplasty [9]. Comparable to our study, the pain intensity was lowest on day 8, with an NSR of 2.3, and increased to 2.6 on day 9, when they were transferred from the acute hospital to the rehabilitation unit.

There are numerous studies that have compared preoperative pain with postoperative pain outcomes. High preoperative knee pain, anxiety, and anticipated pain were the most important predictive factors and had the most influence on satisfaction one year postoperatively [10]. None of the studies found a correlation between preoperative KSS/FS and postoperative pain intensity.

In the future, special attention should be paid to the timing of increasing pain, as the high rate of chronic postoperative pain (CPSP) is alarming. High postoperative pain scores are associated with a higher likelihood of developing CSPS 3 months to a year after the operation [3]. If the pain curve increases by more than 2.8 points, the probability is 33.3% 1 year postoperatively [11]. This study shows the greatest increase in pain on the first day of rehabilitation. However, it confirms that the first few weeks after surgery are the most critical.

### 4.2. Gender of the Patient

When analyzing gender as a possible predictor of postoperative pain intensity, women reported significantly higher pain scores at each surveyed level. This difference is in line with other studies [12,13]. Furthermore, women demonstrated poorer clinical outcomes and lower satisfaction after surgery [14,15]. The gender difference has also been analyzed in other reviews, concluding that women are at increased risk for developing more severe postoperative pain conditions and subsequent CSPS [16]. One reason for this could be because women have more sensory pain fibers [17]. Women report having higher levels of general anxiety as well as factors that capture pain-related stress [16]. Another aspect is that women generally undergo surgery later compared to men and often have greater movement limitations preoperatively [18]. This suggests that earlier treatment in women would improve postoperative outcomes. All in all, several psychosocial, biological, and sociocultural mechanisms may play an important role in the emergence of gender differences in pain.

### 4.3. BMI at Surgery

Another known risk factor for the development of knee osteoarthritis is a high BMI [19,20]. This study population has a total of 91.3% overweight patients (BMI > 30 kg/m^2^). In Germany, 67.1% of men and 53.0% of women are overweight [21]. Therefore, our patients are well above the German average. In the present study, BMI was investigated as a possible influencing factor on the postoperative pain course after TKA. Normal-weight patients reported significantly more severe pain in the postoperative period up to four weeks compared to overweight patients. Here, there was no correlation between gender and BMI. One explanation could be the increased motivation to move and the associated more severe postoperative pain in normal-weight patients. This theory cannot be substantiated in this work. The literature on the effects of BMI on pain and functional outcomes after TKA is somewhat inconsistent. Several studies have shown that the risk of revision after TKA is higher in obese patients than in nonobese patients [22,23]. Chen et al. [23] reported similar clinical outcomes after TKA. Compared with normal-weight patients, obese patients showed significantly higher improvement in the Oxford Knee Score (OKS) and KSS two years after surgery. Another study that evaluated preoperative and 12-month postoperative clinical scores demonstrated greater improvement in overweight patients [24]. High BMI, as well as female gender, Indian/Malay race, and use of general anesthesia compared with regional anesthesia, are identified as influencing factors of “severe pain” [25]. It is important to note that this study did not consider the increased complication rate in obese patients. All in all, a clear benefit of surgery can be obtained regardless of weight.

### 4.4. Age at Surgery

The predictor age was analyzed, and no association was found in this study. Previous work, however, showed partly different results. For example, a patient aged over 70 years showed statistically significantly worse EQ-5D and WOMAC scores [15].

### 4.5. Operation Type

In our evaluation, the type of operation was not a risk factor for more severe pain progression. Preexisting studies also found no clinically important differences between computer-assisted and conventionally performed TKR [26,27,28]. Kim et al. [28] prospectively compared patients who received a computer-navigated knee arthroplasty in one knee and a knee arthroplasty without computer navigation in the other knee. Both groups had similar clinical function, position, and component survival. In contrast, a randomized, double-blind responder analysis showed that more patients with computer-assisted TKR were pain free and had better function after two years than in the conventional group [29].

### 4.6. Perioperative Factors

Surgical time as a possible cause for increased pain intensity was also analyzed, as the duration of the operation may reflect the complexity of the implantation. The repeated resection of bone or the more frequent placement of trial implants during the procedure may influence postoperative pain development. Nevertheless, this hypothesis found no support in the present study report. The same conclusion was also reached in another prospective study [30].

Perioperative blood loss and postoperative pain after TKA could also be issues preventing early mobilization of patients [31]. The effect of tranexamic acid in reducing perioperative blood loss has been described extensively in the past. Several studies have shown a significant reduction in blood loss when using tranexamic acid [32,33].

In the evaluation of the possible influence of the ASA score on the postoperative pain level, no correlation was found.

The duration of the pain catheter was not found to be a possible cause of increased pain intensity. Another study [34] showed that continuous femoral nerve block for at least 72 h resulted in good control of acute postoperative pain as well as early joint mobilization. In the first 24 h after surgery, the 243 patients included reported a VAS of 0-1. All patients achieved 90 degrees of flexion by postoperative day 7. The proximal peripheral nerve block is a commonly used method in pain control after TKA because of its excellent analgesic effect and is considered the gold standard for postoperative analgesia after TKA. However, it may decrease quadriceps strength, which is essential for early mobilization. The adductor canal block might be a reasonable alternative, providing a predominantly sensory block with greater quadriceps strength. [35,36]

The influence of mental health on physical well-being and pain was not investigated in this study, but it also has a major role in postoperative outcome. Anxiety symptoms and depression are likely risk factors for poor outcomes [37]. Similarly, preoperative sleep quality correlates with clinical outcomes (i.e., pain, ROM, function, and length of hospital stay) after total joint arthroplasty [38]. Patients living alone also have a longer hospital stay [39].

### 4.7. Radiological Parameters

About 10% of all TKA patients had a valgus deformity [40]. Valgus of the knee is one of the main reasons for knee joint disease and bears many complications. With this type of deformity, the surgeon must achieve proper alignment, stability, and balance to achieve successful clinical outcomes. The study showed a significant correlation between a valgus leg axis and higher postoperative pain scores. Similar trends could be found in the literature. A study that looked at the factors influencing the prolonged postoperative hospital length of stay noted that preoperative valgus deformity of the knee was a risk factor [41]. Another study compared the postoperative outcomes of valgus and varus leg axes. This showed that patients with a valgus deformity had a WOMAC stiffness score that was significantly worse than the valgus one year postoperatively [42]. Thus, patients with increasing valgus deformity should not wait too long to receive surgical care.

### 4.8. Limitations

The main limitation of the study is the single center setting. Possible important predictors such as psychosocial factors or the radiological severity of knee osteoarthritis were not recorded. The standardized rehabilitation treatment in our rehabilitation facility could be considered a possible selection bias. Also, further information on pain progression, such as pain severity, was not collected at the 3-month follow-up visits. Another limitation is the inclusion of patients during the COVID-19 pandemic. Patients with SARS-CoV-2 infection or contact had to be excluded because they had to stop stationary therapy earlier, both in the acute hospital and in the rehabilitation clinic.

## 5. Conclusions

In this study, the course of pain after total knee arthroplasty showed another peak after nine days. Female gender, low BMI, and preoperative valgus deformity as risk factors resulted in significantly higher postoperative pain scores. This knowledge should be taken into account by surgeons in the future to reduce patient dissatisfaction and prevent chronic pain after primary total knee arthroplasty informing the patient and by counteracting the risk factors at an early stage.

Hereafter, studies should also consider psychological factors, as the perception of pain is individual.

## Figures and Tables

**Figure 1 jcm-11-07204-f001:**
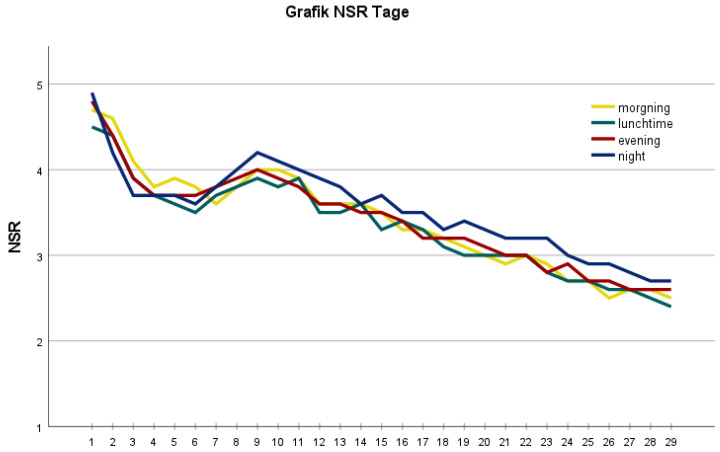
Graph represents the mean pain and maximum pain on days 1–29 on an NRS scale. The mean pain calculation is based on morning, lunchtime, evening, and nighttime pain values.

**Figure 2 jcm-11-07204-f002:**
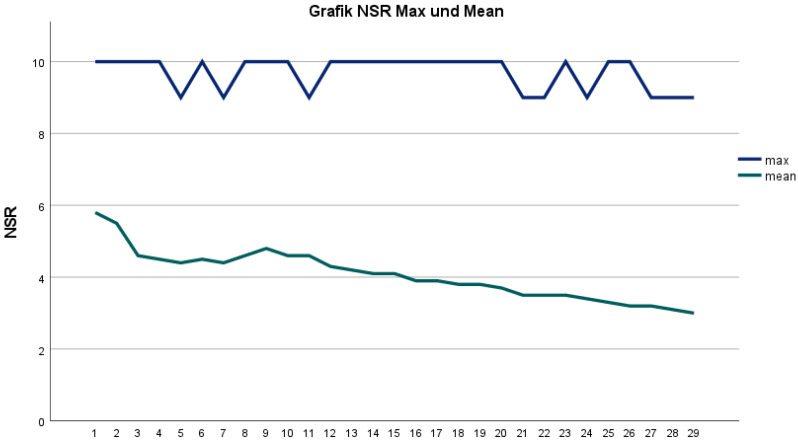
Graph represents the morning, lunchtime, evening, and nighttime pain on days 1–29 on an NRS scale.

**Figure 3 jcm-11-07204-f003:**
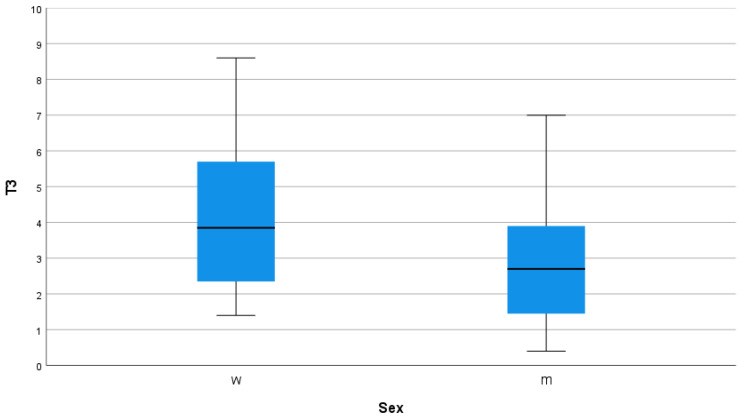
Boxplot over the total analyzed postoperative time and gender (w = women; m = men).

**Table 1 jcm-11-07204-t001:** Linear regression of pain after TKA, based on mean pain levels of the first postoperative week in the acute hospital (T1), the second week until the fourth postoperative week in the rehabilitation unit (T2), and the total analyzed postoperative time (T3); KSS1: knee society score preoperative; FS1: functional score preoperative; KSS2 and FS2 after one week in the acute hospital; and KSS3 and FS2 after four weeks in a rehabilitation unit.

Predictor	B (95% CI)	*p*-Value	*R*^2^ Value
Sex			
T1	−0.999 (−1.84, −0.16)	0.020	−0.257
T2	−1.548 (−2.41, −0.69)	0.001	−0.375
T3	−1.417 (−2.23, −0.61)	0.001	−0.362
Age			
T1	−0.026 (−0.08,0.02)	0.280	−0.118
T2	−0.026 (−0.08, 0.02)	0.290	−0.112
T3	−0.027 (−0.07, 0.02)	0.250	−0.121
BMI			
T1	−0.105 (−0.19, −0.03)	0.010	−0.309
T2	−0.124 (−0.21, −0.04)	0.003	−0.343
T3	−0.120 (−0.20, −0.04)	0.003	−0.349
ASA Score			
T1	0.311 (−0.44,1.06)	0.409	0.089
T2	0.669 (−0.09,1.43)	0.085	0.180
T3	0.589 (−0.13,1.31)	0.110	0.166
Surgical duration			
T1	0.004 (−0.02,0.03)	0.768	0.035
T2	0.017 (−0.01,0.04)	0.166	0.158
T3	0.014 (−0.01,0.04)	0.236	0.134
Paincatheter duration			
T1	0.026 (−0.50, 0.56)	0.922	0.010
T2	0.251 (−0.29, 0.79)	0.359	0.087
T3	0.203 (−0.31,0.71)	0.432	0.075
Previous surgery			
T1	0.037 (−0.85, 0.92)	0.935	0.009
T2	−0.413 (−1.31, 0.49)	0.366	−0.091
T3	−0.303 (−1.16, 0.55)	0.483	−0.071
Operation type			
T1	0.387 (−0.39, 1.17)	0.328	0.097
T2	0.377 (−0.45, 1.21)	0.369	0.089
T3	0.380 (−0.41, 1.17)	0.340	0.095
Anatomical axis 1			
T1	0.065 (0.01, 0.12)	0.020	0.240
T2	0.062 (0.01, 0.12)	0.030	0.215
T3	0.061 (0.01,0.11)	0.022	0.226
Anatomical axis 2			
T1	0.026 (−0.09, 0.14)	0.648	0.044
T2	0.008 (−0.11, 0.12)	0.888	0.013
T3	0.014 (−0.09, 0.12)	0.794	0.024
KSS 1			
T1	0.011 (−0.01,0.04)	0.378	0.086
T2	−0.004 (−0.03, 0.02)	0.733	−0.032
T3	−0.001 (−0.03,0.02)	0.961	−0.005
FS 1			
T1	−0.002 (−0.03, 0.02)	0.897	−0.014
T2	−0.006 (−0.03, 0.02)	0.666	−0.046
T3	−0.005 (−0.03, 0.02)	0.694	−0.042
KSS 2			
T1	−0.027 (−0.06, 0.00)	0.055	−0.231
T2	−0.027 (−0.06, 0.00)	0.064	−0.214
T3	−0.027 (−0.05, 0.00)	0.051	−0.225
FS 2			
T1	−0.012 (−0.05, 0.02)	0.476	−0.073
T2	−0.018 (−0.05, 0.02)	0.292	−0.104
T3	−0.016 (−0.05, 0.02)	0.319	−0.098
KSS 3			
T1	−0.007 (−0.03, 0.02)	0.612	−0.060
T2	−0.008 (−0.04, 0.02)	0.577	−0.064
T3	−0.008 (−0.03, 0.02)	0.549	−0.068
FS 3			
T1	0.009 (−0.03, 0.05)	0.655	0.052
T2	0.022 (−0.02, 0.06)	0.276	0.122
T3	0.019 (−0.02, 0.06)	0.326	0.109

## Data Availability

The datasets generated for this study are available on request to the corresponding author.

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
