# Peer review of "Pain Course after Total Knee Arthroplasty within a Standardized Pain Management Concept: A Prospective Observational Study"

_jcm, 2022, doi:10.3390/jcm11237204_

Round 1

Reviewer 1 Report

This is a well-designed prospective cohort study, investigating the trend of pain after TKA under the standard analgesic scheme.     However, there were some issues that would require addressed:

Abstract

Line 27: BMI is an abbreviation and should be annotated when first mentioned

Introduction

The introduction is well written and clear.

Materials and Methods

Line 100-105: In previous studies, the femoral mechanical axis and tibial mechanical axis were usually used to evaluate the varus or valgus deformity of patients. Femoral anatomic axis and mechanical axis modalities were used in this study for evaluation, please explain the reason.

Results

Line 142-143: A total of 61.2% received a computer-assisted surgery and 38.8% a conventional TKA. Many studies have reported that computer assisted TKA is less traumatic to the soft tissue than conventional TKA, which may cause less pain. This may interfere with the results of the present study, is a subgroup analysis necessary?

Figure 3: Please explain the meaning of "w" and "m".

Discussion

Line 268-278: The authors concluded that valgus deformity was associated with an increased degree of pain. The discussion section only lists similarities and differences with conclusions drawn from other studies, suggesting further analysis of the reasons for the association of valgus with pain extent.

The authors adopted a standard analgesic protocol developed by themselves and ultimately concluded that pain scores would increase during the first postoperative week, possibly attributable to rehabilitative training. This appears to be an expected outcome and cause but does not highlight the implications of this study. Please try to discuss how instructive it is for the clinic to draw a pain curve.

Please cite the following references to discuss the effect of blood management on pain in joint replacement “The Journal of arthroplasty 33 (3), 786-793”.

Potential effect of blood loss on perioperative pain in joint replacement needs to be discussed. Please cite the following references to discuss the effect of blood management on pain in joint replacement “The Journal of arthroplasty 33 (3), 786-793”. “Journal of Thrombosis and Haemostasis 16 (12), 2442-2453”

The perioperative psychological state and sleep of joint replacement patients may have a potential impact on postoperative pain. Please cite the following literature and discuss further. “Journal of Orthopaedic Surgery and Research 14 (1), 1-10” “Orthopaedic Surgery 12 (1), 153-161”

Author Response

Response to Reviewer 1: Thank you very much for your time involved in reviewing the manuscript and your very encouraging comments. We have gone through your comments carefully and tried our best to address them one by one. We hope the manuscript after careful revisions meet your high standards. 

[Comment 1] Abstract: Line 28: BMI is an abbreviation and should be annotated when first mentioned

Response: Thank you very much for the reminder. We have now completed the 
changes you requested. [Page 1, Line 28]

[Comment 2] Materials and Methods: Line 100-105: In previous studies, the femoral mechanical axis and tibial mechanical axis were usually used to evaluate the varus or valgus deformity of patients. Femoral anatomic axis and mechanical axis modalities were used in this study for evaluation, please explain the reason.

Response: Thanks for your question. We thought about this a lot before the radiological measurement and also did research in the literature. It is true that mainly the mechanical axis is used. However, the knee socity score uses the anatomical leg axis, we decided to use this. We hope this explanation has fully addressed all of your concerns. 

[Comment 3] Results: Line 142-143: A total of 61.2% received a computer-assisted surgery and 38.8% a conventional TKA. Many studies have reported that computer assisted TKA is less traumatic to the soft tissue than conventional TKA, which may cause less pain. This may interfere with the results of the present study, is a subgroup analysis necessary?

Response:   Thank you. It’s a good question. We added this remark to the results and discussion. [Table 1, Page 9, Line 268-278]

[Comment 4] Figure 3: Please explain the meaning of "w" and "m".

Response: Thank you for this excellent observation. We have modified Fig.3 as follows and upload it.

[Comment 5] Discussion:  Line 268-278: The authors concluded that valgus deformity was associated with an increased degree of pain. The discussion section only lists similarities and differences with conclusions drawn from other studies, suggesting further analysis of the reasons for the association of valgus with pain extent.

Response:  Unfortunately, we do not understand exactly what modification you want. We have now added important details about the valgus deformity and hope you are satisfied. [Page 9+10, Line 314-318]

[Comment 6] Potential effect of blood loss on perioperative pain in joint replacement needs to be discussed. Please cite the following references to discuss the effect of blood management on pain in joint replacement “The Journal of arthroplasty 33 (3), 786-793”. “Journal of Thrombosis and Haemostasis 16 (12), 2442-2453”

Response: Thank you for your comment. We added this to our discussion. [Page 9, Line 286-291]

[Comment 7] The perioperative psychological state and sleep of joint replacement patients may have a potential impact on postoperative pain. Please cite the following literature and discuss further. “Journal of Orthopaedic Surgery and Research 14 (1), 1-10” “Orthopaedic Surgery 12 (1), 153-161”

Response:  Thank you for underlining this deficiency. Mental health has been completely neglected here and is an important argument. We have re-written this part. [Page 9, Line 306-312].

Reviewer 2 Report

Dear Authors,

I consider your study is original, interesting and well conducted, although there are several concerns:

-       The title should include the characteristics of the study, so it should end with “…: a prospective observational study”

-       The authors should better describe how the peripheral nerve block were performed (ultrasound guide? Neurostimulation? If yes What twitch was observed?

-       The manuscript needs an extensive English revision by a native speaker.

-       What do you mean for proximal ischiatic block? Parasacral? Transgluteal? Subgluteal?

-       What do you mean for “ischiatic nerve block was loaded with ropivacaine 6 mL/h? Was it a continuous block? In this case it should be clearly stated.

-       I noticed you used, as regional anesthesia techniques, proximal peripheral nerve blocks (sciatic nerve and psoas compartment) which are both characterized by sensory and motor block. However, enhanced recovery after surgery (ERAS) protocols recommended the use of local infiltration analgesia or new generation motor sparing fascial block. You should discuss the possibility to use motor sparing regional anesthesia techniques to improve functional outcomes in the postoperative period. At this regards, I suggest you to cite a new promising motor sparing technique aimed to provide a complete analgesia of the surgical area without affecting motor block: the para-sartorial compartments block (The para-sartorial compartments (PASC) block: a new approach to the femoral triangle block for complete analgesia of the anterior knee. Anaesth Rep. 2022)

Author Response

Response to Reviewer 2: We appreciate your clear and detailed feedback and we have made extensive modifications and supplements to our manuscript in order to make our results convincing. We hope this explanation has fully addressed all of your concerns.

[Comment 1] The title should include the characteristics of the study, so it should end with “…: a prospective observational study”

Response: Thank you for pointing out this problem. We have revised it accordingly. 

[Comment 2] The authors should better describe how the peripheral nerve block were performed (ultrasound guide? Neurostimulation? If yes What twitch was observed?

Response:  Thank you for underlining this deficiency. We have re-written this part according to suggestion. [ Page 2, Line 85-87]

[Comment 3] The manuscript needs an extensive English revision by a native speaker.

Response:Thank you for your feedback. A native speaker has now corrected the work. 

[Comment 4] What do you mean for proximal ischiatic block? Parasacral? Transgluteal? Subgluteal?

Response: We have inserted the exact anatomical location. [Page 2, Line 85]

[Comment 5] What do you mean for “ischiatic nerve block was loaded with ropivacaine 6 mL/h? Was it a continuous block? In this case it should be clearly stated.

Response: Sorry for our unclear description. We have now made it more precise. [Page 2, Line 83-95]

[Comment 6]    I noticed you used, as regional anesthesia techniques, proximal peripheral nerve blocks (sciatic nerve and psoas compartment) which are both characterized by sensory and motor block. However, enhanced recovery after surgery (ERAS) protocols recommended the use of local infiltration analgesia or new generation motor sparing fascial block. You should discuss the possibility to use motor sparing regional anesthesia techniques to improve functional outcomes in the postoperative period. At this regards, I suggest you to cite a new promising motor sparing technique aimed to provide a complete analgesia of the surgical area without affecting motor block: the para-sartorial compartments block (The para-sartorial compartments (PASC) block: a new approach to the femoral triangle block for complete analgesia of the anterior knee. Anaesth Rep. 2022)

Response:  Thank you. We added this remark to the discussion. [Page 9, Line 288-294]

Round 2

Reviewer 2 Report

Well done. The overall quality of the manuscript has improved